# A broad-spectrum virus- and host-targeting peptide against respiratory viruses including influenza virus and SARS-CoV-2

Hanjun Zhao[1,2,3,7], Kelvin K. W. To[1,2,3,4,5,7], Kong-Hung Sze[1,2,7], Timothy Tin-Mong Yung[2,3], Mingjie Bian[6], Hoiyan Lam[2,3], Man Lung Yeung[1,2,3,5], Cun Li[2,3], Hin Chu[1,2,3] & Kwok-Yung Yuen [1,2,3,4,5✉]

The 2019 novel respiratory virus (SARS-CoV-2) causes COVID-19 with rapid global socio-economic disruptions and disease burden to healthcare. The COVID-19 and previous emerging virus outbreaks highlight the urgent need for broad-spectrum antivirals. Here, we show that a defensin-like peptide P9R exhibited potent antiviral activity against pH-dependent viruses that require endosomal acidification for virus infection, including the enveloped pandemic A(H1N1)pdm09 virus, avian influenza A(H7N9) virus, coronaviruses (SARS-CoV-2, MERS-CoV and SARS-CoV), and the non-enveloped rhinovirus. P9R can significantly protect mice from lethal challenge by A(H1N1)pdm09 virus and shows low possibility to cause drug-resistant virus. Mechanistic studies indicate that the antiviral activity of P9R depends on the direct binding to viruses and the inhibition of virus-host endosomal acidification, which provides a proof of concept that virus-binding alkaline peptides can broadly inhibit pH-dependent viruses. These results suggest that the dual-functional virus- and host-targeting P9R can be a promising candidate for combating pH-dependent respiratory viruses.

[1] Li Ka Shing Faculty of Medicine, State Key Laboratory of Emerging Infectious Diseases, The University of Hong Kong, Pokfulam, Hong Kong Special Administrative Region, China. [2] Department of Microbiology, Li Ka Shing Faculty of Medicine, The University of Hong Kong, Pokfulam, Hong Kong Special Administrative Region, China. [3] Centre for Virology, Vaccinology and Therapeutics, Health@InnoHK, The University of Hong Kong, Hong Kong, China. [4] Li Ka Shing Faculty of Medicine, Carol Yu Centre for Infection, The University of Hong Kong, Pokfulam, Hong Kong Special Administrative Region, China. [5] The University of Hong Kong-Shenzhen Hospital, Shenzhen, Guangdong Province, China. [6] School of Life Science, Anhui Normal University, Wuhu, Anhui, China. [7] These authors contributed equally: Hanjun Zhao, Kelvin K. W. To, Kong-Hung Sze ✉email: kyyuen@hku.hk

Novel respiratory viruses often cause severe respiratory tract infections and spread quickly due to the lack of pre-existing immunity. In the recent two decades, three highly pathogenic coronaviruses have crossed species barrier and caused human diseases, including the bat-related severe acute respiratory syndrome coronavirus (SARS-CoV) in 2003[1,2], the Middle East respiratory syndrome coronavirus (MERS-CoV) since 2012[3,4] and the recent 2019 new coronavirus (SARS-CoV-2)[5]. Besides, the 2009 pandemic influenza A(H1N1)pdm09 virus had led to the first influenza pandemic in the 21st century, and the avian influenza virus A(H7N9) had caused a large zoonotic outbreak in mainland China[6,7]. Due to the lack of effective antivirals, especially for coronaviruses, these respiratory viruses are associated with significantly high morbidity and mortality. Furthermore, these emerging respiratory viruses have also caused severe economic and social disturbances.

The COVID-19 outbreak has clearly illustrated the importance of broad-spectrum antivirals. While an outbreak of unusual pneumonia was reported from Wuhan of China in December 2019, SARS-CoV-2 was identified on 8 January 2020 by China CDC[8]. Currently, there is no specific drug treatment for this new virus. An effective broad-spectrum antiviral will improve patients' outcome and may reduce transmission in communities and hospitals even before the identification of the novel emerging virus and the specific antiviral drug.

In past decades, researchers have devoted to the discovery of broad-spectrum antivirals targeting hosts or viruses[9,10]. Most broad-spectrum antivirals are targeting host pathways which are utilized by virus replication cycle or by host antiviral immunity[11,12]. We have previously reported a broad-spectrum antiviral peptide P9[13], derived from mouse β-defensin-4, which showed antiviral activity against SARS-CoV, MERS-CoV, and influenza viruses through the inhibition of endosomal acidification. We have also reported different types of antiviral peptides which inhibit influenza virus by delivering defective interfering gene[14], zika virus by targeting envelope protein[15], MERS-CoV by blocking viral fusion[16]. Other groups have reported the broad-spectrum antiviral peptide BanLec which inhibits HIV, hepatitis C virus (HCV) and influenza virus[17], the theta-defensin retrocylin 2, which inhibits influenza virus, Sindbis virus, and baculovirus[18], the typical cationic and amphipathic tetradecapeptide mastoparan which inhibits enveloped viruses[19], two scorpion venom peptide variants including mucroporin-M1 which inhibits measles, SARS-CoV and H5N1 virus[20], and the Ev37 which inhibits dengue virus, HCV, zika virus, and herpes simplex virus[21], and other host defense peptides which inhibit different enveloped and non-enveloped viruses with documented or unknown mechanism[22–24]. Since the first anti-HIV peptide enfuvirtide was approved by FDA for the treatment of HIV in 2003[25], other peptides with antiviral activity have been approved by FDA[26].

Here, we report a potent antiviral peptide P9R, derived from mouse β-defensin-4 and our previous P9[13], which exhibits broad-spectrum antiviral activities against the enveloped SARS-CoV-2, MERS-CoV, SARS-CoV, A(H1N1)pdm09, A(H7N9) virus, and the non-enveloped rhinovirus. In vivo study indicates that P9R can efficiently protect mice from lethal A(H1N1)pdm09 virus challenge. Furthermore, the antiviral resistance study indicates that P9R does not induce the emergence of drug-resistant virus even after A(H1N1)pdm09 virus was passaged in the presence of P9R for 40 passages. Mechanistic studies show that positively charged P9R broadly inhibits viral replication by binding to different viruses and then inhibits virus–host endosomal acidification to prevent the endosomal release of pH-dependent viruses. We use P9R (not only binding to viruses but also inhibiting endosomal acidification), PA1 (only binding to viruses) and P9RS (only inhibiting endosomal acidification) to identify and confirm the antiviral mechanism of alkaline peptides. The antiviral activity of alkaline peptide, requiring both of binding to viruses and inhibition of endosomal acidification, can be improved by increasing the positive charge of the peptide.

## Results

**P9R can broadly inhibit coronaviruses and other respiratory viruses**. Endosomal acidification is affected by the influx of protons into the endosome via the vacuolar membrane proton pump V-ATPase[27]. Theoretically, an alkaline peptide with stronger net positive charge would reduce protons in the endosome, thereby inhibiting the endosomal acidification. Hence, to improve our previously antiviral peptide P9[13], we substituted the weakly positively charged amino acids (histidine and lysine) by arginine at positions 21 (H → R), 23 (K → R) and 28 (K → R) (Fig. 1a) to increase the net positive charge (+4.7) of P9 to charge (+5.6) of P9R. By performing a plaque reduction assay, we showed that the IC$_{50}$ of P9R against SARS-CoV-2 was significantly lower than that of P9 (0.9 μg ml$^{-1}$ vs 2.4 μg ml$^{-1}$, $P < 0.01$) (Fig. 1b). When P9R was added to cells after SARS-CoV-2 infection, P9R could also significantly inhibit viral replication (Supplementary Fig. 1). Furthermore, P9R showed significantly stronger inhibition against MERS-CoV, A(H1N1)pdm09 virus, A(H7N9) virus, and rhinovirus than P9 (Fig. 1c–g). However, the IC$_{50}$ of P9R and P9 against parainfluenza virus 3 was much higher (>25.0 μg ml$^{-1}$), likely because endosomal acidification was not required in the viral life cycle of parainfluenza virus 3 (Fig. 1h)[28]. In the multicycle growth assay, P9R inhibited more than 100-fold viral replication for SARS-CoV-2, MERS-CoV, and SARS-CoV (Fig. 1i). For A(H1N1)pdm09 virus, A(H7N9) virus and rhinovirus, P9R could inhibit >20-fold viral replication (Fig. 1i). In addition, the CC$_{50}$ of P9R was >300 μg ml$^{-1}$ for MDCK, VeroE6 and A549 cells (Fig. 1j). These results indicated that P9R with more positive charge could more efficiently inhibit the new coronavirus SARS-CoV-2 and other enveloped and non-enveloped respiratory viruses than that of P9.

**The positive charge of P9R is critical for the antiviral activity**. To determine whether the net charge of the peptide affects the inhibition of endosomal acidification, we showed that P9R (+5.6) could more significantly inhibit endosomal acidification in live cells than that of P9 (Fig. 2a, b), which was consistent with the stronger antiviral activity of P9R when comparing with P9. In addition, peptide PA1 with less positive charge (+1.7), which has the same amino acid sequence as P9 except 3 additional acidic amino acid at the C terminal, could not inhibit endosomal acidification (Fig. 2a, b) and lacked the antiviral activity (Fig. 2c). Hence, the degree of net positive charge was correlated with the degree of inhibition of endosomal acidification and antiviral activity.

In order to investigate whether P9R could have antiviral action before endosomal acidification, we showed that P9R did not disrupt viral particles under TEM analysis (Supplementary Fig. 2), did not cause hemolysis of RBC (Supplementary Fig. 3) and did not show antiviral activity when cells were pretreated by P9R before viral infection (Supplementary Fig. 4). Furthermore, P9R did not affect viral attachment (Supplementary Fig. 5) and did not inhibit viral fusion by hemolysis inhibition assay (Supplementary Fig. 6). The antiviral activity of P9R was irreversible after binding to virus (Supplementary Fig. 7). Results indicated that the irreversible antiviral activity of P9R did not rely on disrupting lipid membrane or inhibit virus–host fusion by directly binding.

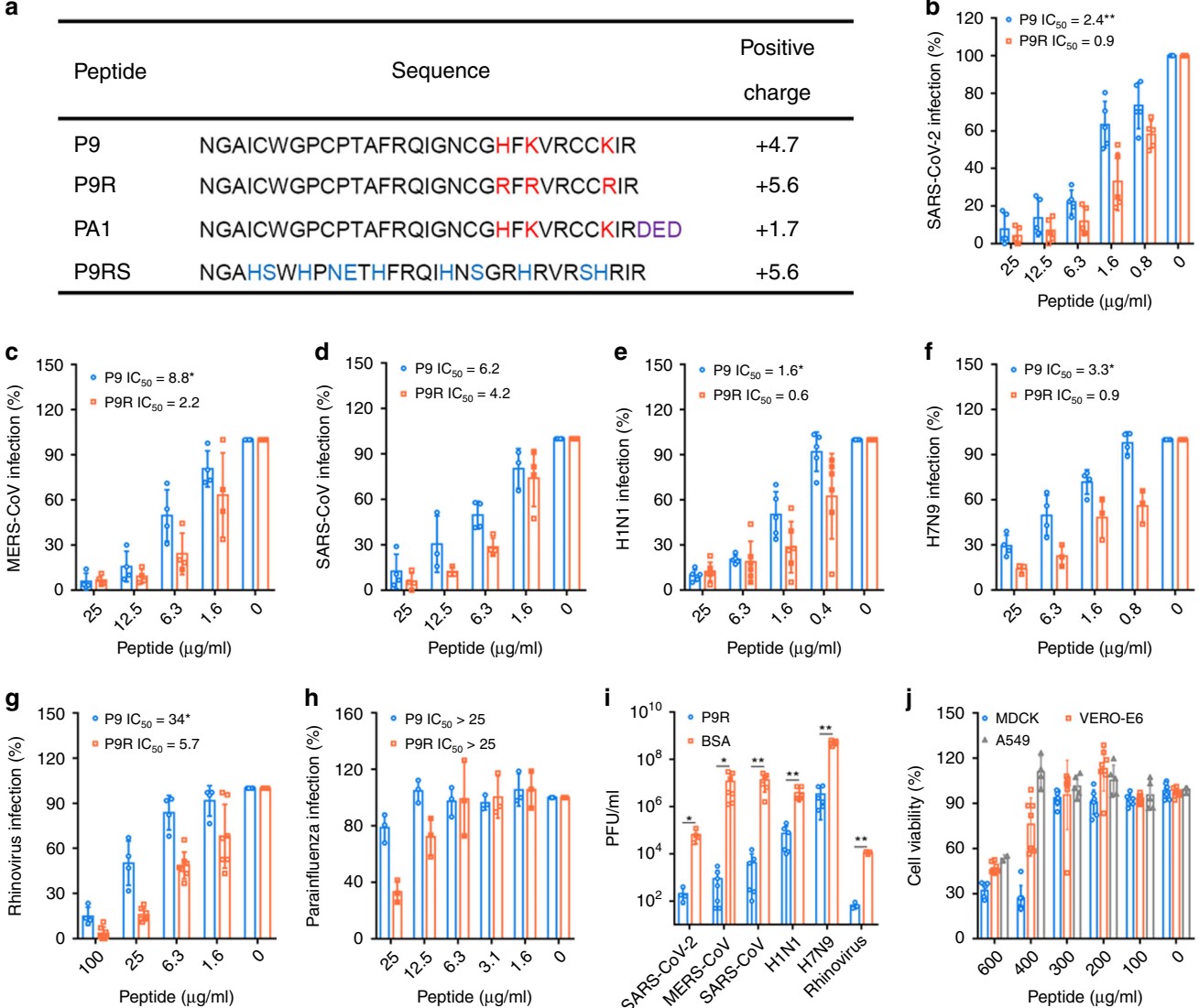

**Fig. 1 P9R could broadly inhibit enveloped and non-enveloped virus replication. a** Peptide sequences and positive charge analyzed by PepCalc of InnovaGen. The net positive charges of peptides are related to pH 7.0. **b–h** P9R could inhibit viral replication of SARS-CoV-2 ($n = 5$), MERS-CoV ($n = 4$), SARS-CoV ($n = 4$), H1N1 virus ($n = 5$), H7N9 virus ($n = 4$), rhinovirus ($n = 4$), but not parainfluenza 3 virus ($n = 3$) in cells. Viruses were premixed with different concentrations of P9R or P9 and then infected cells. The antiviral efficiency was evaluated by plaque reduction assay. Infection (%) was calculated by the plaque number of virus treated with peptides divided by the plaque number of virus treated by BSA. $P = 0.006$ for SARS-CoV-2, $P = 0.042$ for MERS-CoV, $P = 0.02$ for H1N1 virus, $P = 0.03$ for H7N9 virus, and $P = 0.007$ for rhinovirus when $IC_{50}$ of P9R against virus was compared with that of P9. **i** The potent antiviral activities of P9R against viruses including SARS-CoV-2 ($n = 4$, $P = 0.015$), MERS-CoV ($n = 7$, $P = 0.027$), SARS-CoV ($n = 6$, $P = 0.005$), H1N1 ($n = 6$, $P = 0.002$), H7N9 ($n = 6$, $P < 0.001$) and rhinovirus ($n = 3$, $P < 0.001$) by measuring the viral titers in supernatants at 24 h post infection when viruses were treated by P9R or BSA (50–100 μg ml$^{-1}$). Asterisk indicates $P < 0.05$ and double asterisk indicates $P < 0.01$ when compared with virus treated by BSA. $P$ values were calculated by the two-tailed Student's $t$ test. **j** Cytotoxicity of P9R in MDCK ($n = 7$), VERO-E6 ($n = 7$) and A549 ($n = 5$) cells. Data are presented as mean ± SD of at least three independent experiments. Source data are provided as a Source data file.

**Positive charge is not solely responsible for antiviral activity.** To determine whether the antiviral activity solely relied on the positive charge of peptide, we designed a peptide P9RS (+5.6) which had the same positive charge as P9R (+5.6), but P9RS differed from P9R by 11 of 30 amino acids. P9RS efficiently inhibited host endosomal acidification to the similar degree as P9R in live cells (Fig. 2a, b). However, in the plaque reduction assay, there was no significant reduction of plaque numbers for SARS-CoV-2 and A(H1N1)pdm09 virus when viruses were treated by P9RS even at 25 μg ml$^{-1}$ (Fig. 2c).

To investigate why P9RS failed to inhibit viral replication despite potent inhibition of host endosomal acidification, we studied the binding between the peptide and virus. Using ELISA

and RT-qPCR assay, P9R and PA1 could efficiently bind to SARS-CoV-2 and A(H1N1)pdm09 virus but P9RS did not bind to SARS-CoV-2 and A(H1N1)pdm09 virus (Fig. 2d). The observation of P9R but not P9RS binding to virus was further confirmed by confocal microscopy in H1N1-infected cells (Fig. 2e). Thus, the direct interaction of peptide with virus was required for the antiviral activity of positively charged peptide P9R. In contrast, P9RS without the ability of binding to virus could not inhibit viral replication even though it carries the same positive charge as P9R and inhibits host endosomal acidification.

**P9R targets virus to inhibit virus–host endosomal acidification.** In the above experiments, we had demonstrated that P9R and

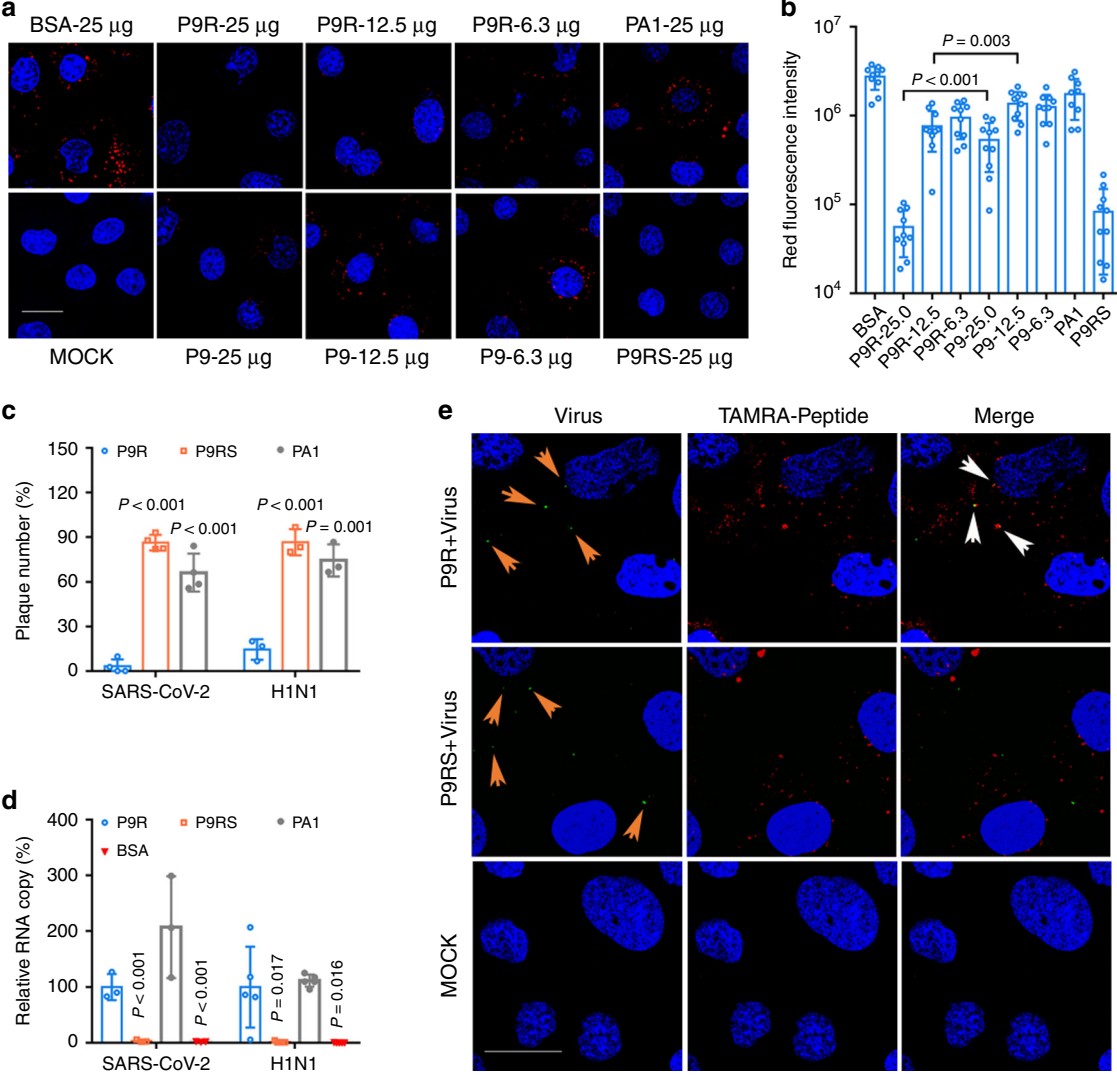

**Fig. 2 The enhanced antiviral mechanism of P9R. a** P9R more efficiently blocked the endosomal acidification than that of P9. MDCK cells were treated by peptide and low pH indicator pHrodo™ dextran. Red dots (pHrodo™ dextran) indicated low pH in endosomes. Blue color indicates nuclei. Live cell images were taken by confocal microscope (scale bar = 20 μm). Experiments were repeated with three biological samples. **b** The quantification of red fluorescence of endosomal acidification in MDCK cells treated by peptides. The red fluorescence intensity was calculated from 10 random microscope fields. **c** Antiviral activities of 25 μg ml$^{-1}$ of P9R, P9RS, and PA1 against SARS-CoV-2 ($n = 4$) and A(H1N1)pdm09 virus ($n = 3$) were measured by plaque reduction assay. Plaque number (%) of peptide-treated virus was normalized to BSA-treated virus. **d** P9R and PA1 could bind to SARS-CoV-2 ($n = 3$) and A(H1N1)pdm09 virus ($n = 5$). Viruses binding to peptides were detected by ELISA and RT-qPCR. $P$ values were calculated by the two-tailed Student's t test when compared with P9R. Data are presented as mean ± SD from at least three independent experiments. **e** P9R could bind to virus in infected cells. Fluorescence labeled virus (green) was pretreated by TAMRA-labeled P9R (red) or TAMRA-labeled P9RS (red) for 1 h. MDCK cells were infected with the treated virus. Images were taken by confocal microscope (scale bar = 20 μm). Experiments were repeated with three biological samples. Orange arrows indicated viral particles. White arrows indicated P9R binding to virus. P9RS did not show binding to virus. Source data are provided as a Source Data file.

P9RS can inhibit the acidification of the no-virus endosomes (Fig. 2a, b). However, without binding to virus, P9RS could not inhibit viral replication. To illustrate this result, we showed that P9R and bafilomycin A1 could efficiently inhibit the virus–host endosomal acidification in infected live cells, but P9RS could not inhibit the virus–host endosomal acidification in infected live cells (Fig. 3a), even though both of P9R and P9RS could inhibit the endosomal acidification of no-virus endosomes (Fig. 2a). The efficient inhibition of P9R on virus–host endosomal acidification could be due to the binding of P9R to virus (Fig. 2d, e) and then inhibiting the virus–host endosomal acidification (Fig. 3a). Lacking the binding ability to viruses (Fig. 2d, e), P9RS could not efficiently enter endosomes with the viruses to inhibit the virus–host endosomal acidification. However, without viruses in endosomes, P9RS freely entered endosomes to prevent endosomal acidification (Fig. 2a). It should be noted that PA1 with a similar sequence as P9R could efficiently bind to SARS-CoV-2 and A (H1N1)pdm09 virus (Fig. 2d), but it had significantly less antiviral activity against SARS-CoV-2 and A(H1N1)pdm09 virus (Fig. 2c). The binding of P9R to SARS-CoV-2 and A(H1N1) pdm09 virus could be significantly reduced when viruses were pretreated by PA1 (Fig. 3b). This indicated that PA1 had the same binding sites on viral particles as P9R. It also suggested that only peptide binding to virus alone could not account for the antiviral activity. P9R binding to virus was the first step to exert the antiviral activity. After binding to virus (Fig. 2e), P9R could efficiently inhibit virus–host endosomal acidification (Fig. 3a) and then inhibit viral replication by blocking RNP release (Fig. 3c).

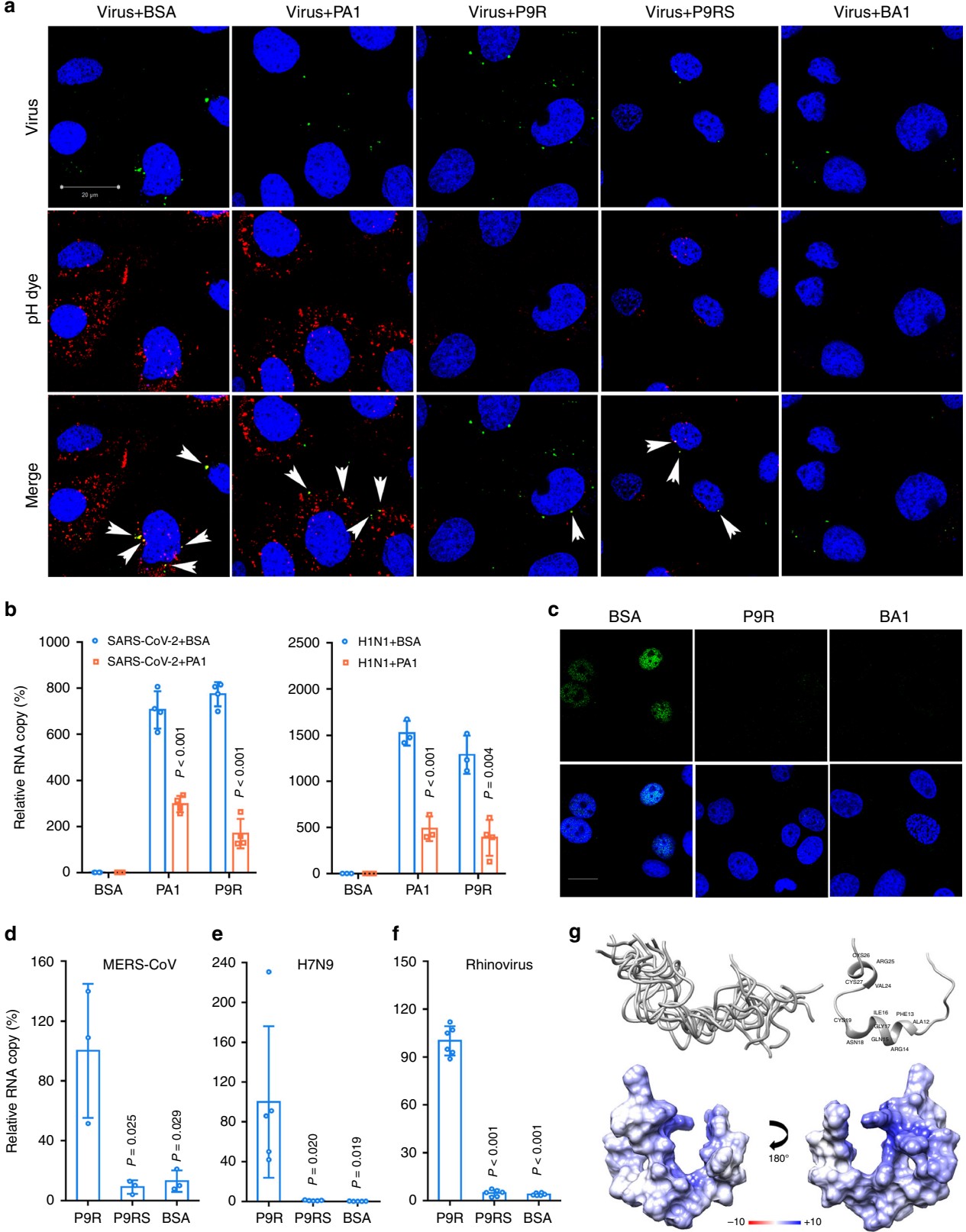

To further confirm that the broad-spectrum antiviral activity of P9R was due to the broad spectrum of binding of P9R to different viruses and viral proteins, we demonstrated that P9R but not P9RS could also bind to MERS-CoV, A(H7N9) virus, rhinovirus, SARS-CoV and viral proteins (Fig. 3d–f and Supplementary Fig. 8). This result further confirmed that positively charged P9R

could inhibit pH-dependent endosomal viruses if it can bind to viruses.

Next, we determined the structure of P9R using NMR spectroscopy (Fig. 3g and Supplementary Table 1). The results indicated that the solution structure of P9R was flexible with two local ordered regions (residues 8–17 and 22–29) containing short

**Fig. 3 P9R could inhibit virus–host endosome acidification through directly binding to viruses. a** Fluorescence labeled virus (green) was pretreated with indicated peptides (25 μg ml$^{-1}$) or bafilomycin A1 (BA1). MDCK cells were infected with the treated H1N1-virus and low pH sensitive dye (red) was added to cells. Live cell images were taken by confocal microscope (scale bar = 20 μm). Experiments were repeated with three biological samples. Red dot indicated the endosomal acidification. Nuclei was stained as blue. White arrows indicated the colocalization of virus and endosomal acidification. **b** P9R binding to SARS-CoV-2 ($n = 4$) and A(H1N1)pdm09 ($n = 3$) could be reduced by PA1. Virus was pretreated by PA1 or BSA, and then the treated virus binding to the indicated peptides were measured by RT-qPCR. $P$ values were calculated by the two-tailed Student's t test when compared with virus treated by BSA. **c** P9R could inhibit viral RNP release into nuclei. H1N1 virus was pretreated by BSA, P9R or bafilomycin A1 (BA1), and then MDCK cells were infected with the treated virus. Images of viral NP (green) and cell nuclei (blue) were taken at 3.5 h post infection. Scale bar = 20 μm. Experiments were repeated with three biological samples. **d–f** P9R could broadly bind to MERS-CoV ($n = 3$), H7N9 virus ($n = 5$), and rhinovirus ($n = 6$). The relative RNA copy of virus binding to peptides was normalized to the virus binding to P9R. $P$ values were calculated by the two-tailed Student's t test when compared with P9R. Data are presented as mean ± SD from at least three independent experiments. **g** Top left: top 10 water-refined structures of P9R generated by ARIA displayed as backbone traces, showing an overall flexible structure. Top right: representative NMR structure of P9R displayed as ribbon plot showing the formation of some helix patches. Bottom: surface charge plot of the representative NMR structure of above P9R showing largely positively charged peptide surface. Source data are provided as a Source data file.

variable helical patches and with positively charged peptide surface (Fig. 3g). We hypothesized that P9R could broadly bind to different viruses because these short α-helical patches with flexible linkages may allow it to adapt its structure to fit the binding pockets of different viral proteins. More co-binding structure analysis will be needed for identifying the broadly binding mechanism of P9R with different viral proteins in future. In conclusion, here we demonstrated the antiviral mechanism that positively charged P9R binds target viruses and then prevents virus–host endosomal acidification to inhibit pH-dependent virus replication cycles.

**The efficacy of P9R treatment in vivo.** We demonstrated that the efficient antiviral activity of P9R in vitro is reliant on binding to viruses and the positive charge of P9R to inhibit virus–host endosomal acidification. To further investigate the antiviral activity of P9R in vivo, we treated A(H1N1)pdm09-infected mice at 6 h post infection and gave additional two doses to mice in the following one day. In this model, 70% of P9R-treated mice and 80% of zanamivir-treated mice survived, which was significantly better than PBS-treated group and PA1-treated group (Fig. 4a). The protection of P9R (70%) on infected mice was better than P9 (50%). From day 4 to day 6 post infection, there was significantly less body weight loss of mice in P9R group than that in PBS-treated group and PA1-treated group (Fig. 4b). P9R and zanamivir could significantly inhibit viral replication in mouse lungs when compared with that of PBS-treated mice (Fig. 4c) The better survival, the reduced weight loss and low viral loads of mice treated by P9R (25 μg/dose and 12.5 μg/dose) further demonstrated that P9R could significantly protect mice when compared with PBS-treated mice (Fig. 4d–f). The survival of P9R in vivo was better than that of P9 (Fig. 4c, $P < 0.05$ for 12.5μg/dose), which was consistent with the significantly better antiviral activity of P9R than P9 in vitro.

**No emergence of resistant viruses against P9R after serial passages.** Emergence of resistant mutants occur from time to time[14], especially with the new polymerase inhibitor baloxavir[29]. To determine whether P9R treatment induces viral resistance, A(H1N1)pdm09 virus was serially passaged 40 times in the presence of P9R in MDCK cells (Fig. 5a). A(H1N1)pdm09 virus was serially passaged in the presence of zanamivir as a control for resistance assay (Fig. 5a). The IC$_{50}$ of zanamivir against parent A(H1N1) pdm09 virus (P0) was 35 nM (Fig. 5b). After 10-virus passages in the presence of zanamivir (100 nM) and additional 5-virus passages in the presence of zanamivir (1000 nM), 2000 nM and 8000 nM zanamivir could not inhibit P10 and P15 virus replication, respectively (Fig. 5c). These indicated that after

10 passages of virus in the presence of zanamivir had caused significant viral resistance to zanamivir. However, for P9R, even the A (H1N1)pdm09 virus was passaged in the presence (5.0 μg ml$^{-1}$ of P9R for the initial 10 passages and 50.0 μg ml$^{-1}$ for the rest 30 passages) of P9R for 40 passages, P9R (5.0 μg ml$^{-1}$) could efficiently inhibit P30 and P40 virus replication (Fig. 5d). No obvious drug-resistant virus to P9R was detected. These results indicated that P9R had very low possibility to cause drug-resistant virus.

**Discussion**

In this study, we identified a broad-spectrum antiviral peptide P9R with potent antiviral activity against enveloped coronaviruses (SARS-CoV-2, SARS-CoV, and MERS-CoV), influenza virus, and non-enveloped rhinovirus. First, we demonstrated that the antiviral activity of P9R could be significantly enhanced by increasing the net positive charge for more efficient inhibition of endosomal acidification. Second, antiviral mechanistic studies further demonstrated that positively charged P9R could bind to different respiratory viruses to inhibit virus–host endosomal acidification, while PA1 (only binding to viruses) or P9RS (only inhibiting endosomal acidification) did not show antiviral activity. Third, the in vivo antiviral activity of P9R was demonstrated by protecting mice from lethal influenza virus challenge. Fourth, there was no reduced susceptibility of serially passaged viruses (40 passages) against P9R.

The "one bug-one drug" approach to antiviral drug is successful for HIV, hepatitis C virus and influenza virus[9]. However, there is an urgent need for broad-spectrum antivirals for combating emerging and re-emerging new virus outbreaks, such as the SARS-CoV-2, before the new virus is identified or specific antiviral drug is available.

Endosomal acidification is a key step in the life cycle of many pH-dependent viruses, which is one of the broad-spectrum antiviral targets[9]. In this study, with the increased positive charge in P9R, it could more efficiently inhibit pH-dependent viruses than that of P9. The more positive charge in P9R allowed the peptide to more efficiently reduce protons inside endosomes, and thereby inhibiting the endosomal acidification. In previous studies, the clinically approved anti-malarial drug chloroquine and hydroxychloroquine with activity of inhibiting endosomal acidification had been demonstrated to inhibit enterovirus-A71[30], zika virus[31] and SARS-CoV-2[32,33]. The anti-parasitic drug niclosamide also inhibited influenza virus, rhinovirus, and dengue virus by interfering endosomal acidification[34,35]. However, researchers demonstrated the lack of protection of chloroquine in vivo for treating influenza virus and Ebola virus[36,37]. Differing from these drugs by interfering host endosomal acidification without targeting viruses, P9R inhibits viral replication by binding to viruses and then inhibiting host acidification of the

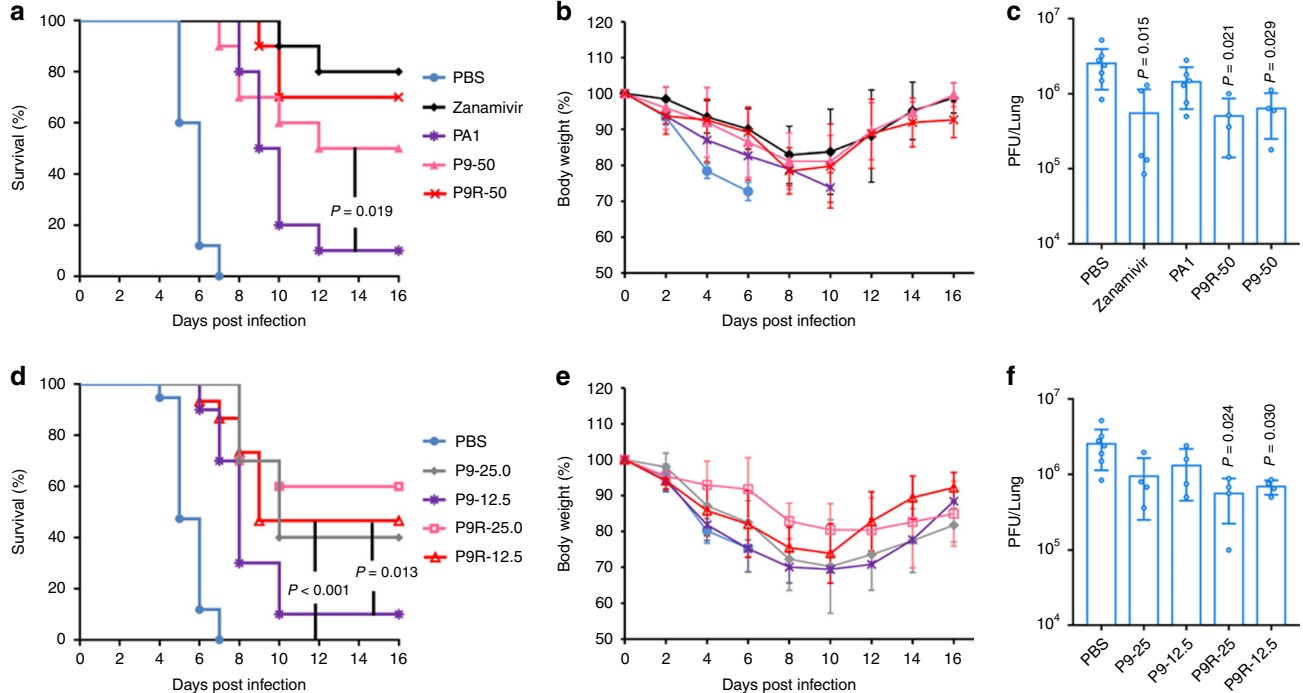

**Fig. 4 The therapeutic protection of P9R on A(H1N1)-infected mice. a** P9R (50 μg/dose) could provide similar therapeutic efficacy on mice infected by A (H1N1) virus as that of zanamivir (50 μg/dose). PBS (n = 10), zanamivir (n = 10), PA1 (n = 10), P9R (n = 10), or P9 (n = 10) were intranasally inoculated to mice at 6 h post infection and two more doses were administrated to mice in the following one day. P value was calculated by Gehan–Breslow–Wilcoxon test. **b** The body weight change of infected mice corresponding to **a**. **c** Viral loads in lung tissues corresponding to **a**. **d** Low doses of P9R could more efficiently protect mice infected by A(H1N1)pdm09 virus than that of P9. PBS (n = 20), P9-25 (25.0 μg/dose, n = 10), P9-12.5 (12.5 μg/dose, n = 10), P9R-25 (25.0 μg/dose, n = 10), and P9R-12.5 (12.5 μg/dose, n = 15) were intranasally inoculated to mice at 6 h post infection and two more doses were administrated to mice in the following one day. P value was calculated by Gehan–Breslow–Wilcoxon test. **e** The body weight change of infected mice corresponding to **d**. **f** Viral loads in lung tissues corresponding to **d**. Ten to twenty mice in each group for survivals were included. P values of survivals were calculated by Gehan–Breslow-Wilcoxon test. For viral loads in mouse lungs, data are presented as mean ± SD from more than three mice in each group. P values of viral loads were calculated by the two-tailed Student's t test when compared with mice treated by PBS. Source data are provided as a Source data file.

endosomes containing the virus bounded by P9R, which allows P9R to selectively and efficiently inhibit the replication cycle of endosomal viruses. The protection of P9R on A(H1N1)-infected mice further confirmed the antiviral efficiency in vivo.

The antiviral activity of P9R required both the binding to viruses and the inhibition of endosomal acidification. PA1 with less positive charge could not inhibit SARS-CoV-2 and H1N1 virus even though it had the similar binding ability and binding sites to viruses as P9R (Fig. 3b). When we made multiple substitutions on P9R to generate P9RS, P9RS lost the binding ability and antiviral activity to all tested viruses even though P9RS has the same positive charge as P9R and retains the ability to inhibit host endosomal acidification. The broadly binding mechanism of P9R to different viral proteins may be due to the flexible structure of P9R with positively charged surface (Fig. 3g). The flexible structure may allow P9R to change its structure to fit targeting proteins for broad-specificity binding[38,39], and the positive charge of P9R may play a role for binding to viruses with negatively charged surface[40,41]. The five cysteines in P9R may also affect the structure-based binding because previous studies indicated that cysteine substitutions could affect defensin-peptide structure and biological activity[42,43].

In addition, comparing with zanamivir which caused significant drug-resistant virus after 10-virus passages in the presence of zanamivir, P9R showed very low possibility to cause drug-resistant virus even A(H1N1)pdm09 virus was passaged in the presence of P9R for 40 passages. The low risk of resistance induction by P9R may be partially due to the dual targeting

abilities which broadly bind to viruses and target host endosomes. More co-binding structure analysis will be needed to illustrate the broadly binding mechanism and help to understand the low drug resistance mechanism in future study.

In summary, most highly pathogenic emerging viruses are endosomal pH-dependent viruses. The emerging and re-emerging virus outbreaks remind us of the urgent need of broad-spectrum antivirals. Continual studies on similar kind of antivirals, which can broadly bind to different viruses and inhibit pH-dependent viruses by preventing virus–host endosomal acidification with low possibility of inducing drug resistance, will give us more armamentarium to combat novel emerging virus outbreaks in future.

## Methods

**Cells and virus culture**. Madin Darby canine kidney (MDCK, CCL-34), Vero-E6 (CRL-1586), RD (CCL136), LLC-MK2 (CCL-7), A549 (CCL-185) cells obtained from ATCC (Manassas, VA, USA) were cultured in Dulbecco minimal essential medium (DMEM) or MEM supplemented with 10% fetal bovine serum (FBS), 100 IU ml$^{-1}$ penicillin and 100 μg ml$^{-1}$ streptomycin. The virus strains used in this study included 2019 new coronavirus (SARS-CoV-2)[44], SARS-CoV, MERS-CoV (hCoV-EMC/2012), A/Hong Kong/415742/2009, A/Hong Kong/415742Md/2009 (H1N1) (a highly virulent mouse-adapted strain), A/Anhui/1/2013 (H7N9)[13], rhinovirus[45] and human parainfluenza 3 (ATCC-C243). For in vitro experiments, viruses were cultured in MDCK, VERO-E6, RD and LLC-MK2 cells. For animal experiments, H1N1 virus was cultured in eggs[46].

**Design and synthesis of peptides**. P9, P9R, PA1 and P9RS were designed as shown in Fig. 1a and synthesized by ChinaPeptide (Shanghai, China). All peptides were dissolved in water. The solubility of P9R in water is >5 mg/ml. The purity of

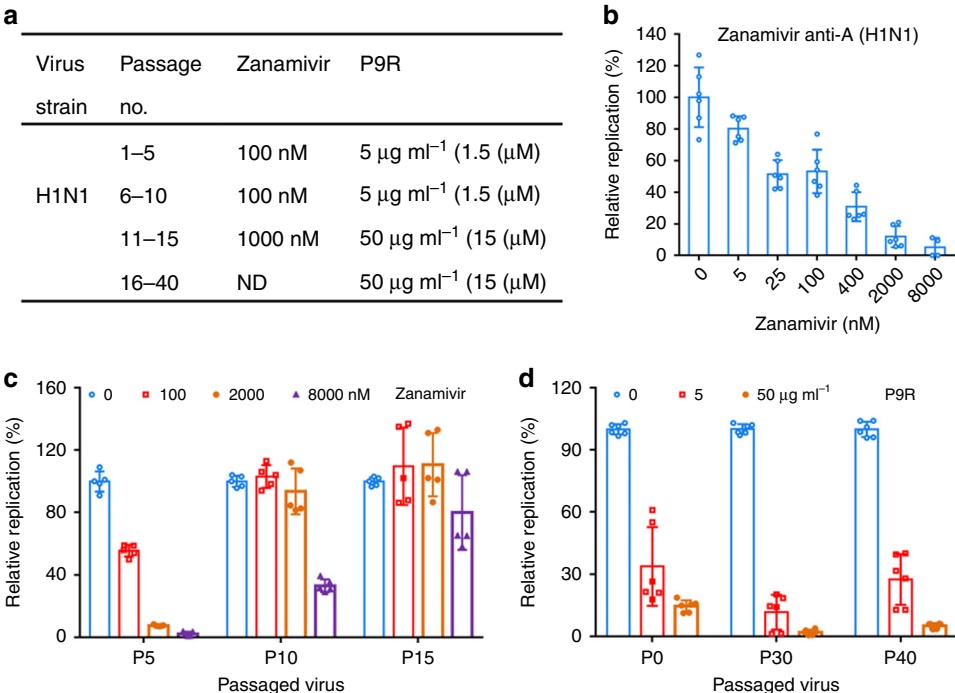

**Fig. 5 P9R showed low possibility to cause drug-resistance virus. a** The procedure of drug-resistance assay for zanamivir and P9R. A(H1N1) virus was passaged in the presence of indicated concentrations of zanamivir or P9R. ND, not detected because the high resistant H1N1 virus against zanamivir was generated before P16. **b** Zanamivir inhibited parent A(H1N1) virus (P0). The IC$_{50}$ of zanamivir against parent H1N1 was 35 nM. Data are presented as mean ± SD of three independent experiments. **c** The antiviral efficiency of zanamivir against passaged A(H1N1) virus in the presence of zanamivir ($n = 5$). **d** The antiviral efficiency of P9R against passaged A(H1N1) virus in the presence of P9R ($n = 6$). Passaged viruses were premixed with zanamivir (nM) or P9R (µg ml$^{-1}$) for infection. Supernatants were collected at 24 h post infection. Viral titers in the supernatants were determined by RT-qPCR. The relative replication (%) was normalized to the corresponding passaged viruses without treatment. Data are presented as mean ± SD of at least three independent experiments. Source data are provided as a Source data file.

all peptides was >95%. The purity and mass of each peptide were verified by HPLC and mass spectrometry.

**Plaque reduction assay**. Antiviral activity of peptides was measured using a plaque reduction assay[14]. Briefly, peptides were dissolved in 30 mM phosphate buffer (PB) containing 24.6 mM Na$_2$HPO$_4$ and 5.6 mM KH$_2$PO$_4$ at a pH of 7.4. Peptides or bovine serum albumin (BSA, 0.4–50.0 µg ml$^{-1}$) were premixed with 50 PFU of coronaviruses (SARS-CoV-2, MERS-CoV, and SARS-CoV), influenza viruses (H1N1 virus and H7N9 virus), rhinovirus, or parainfluenza 3 in PB at room temperature. After 45–60 min of incubation, peptide-virus mixture was transferred to VERO-E6, MDCK, RD, or LLC-MK2 cells, correspondingly. At 1 h post infection, infectious media were removed and 1% low melting agar was added to cells. Cells were fixed using 4% formalin at 2–4 day post infection. Crystal blue (0.1%) was added for staining, and the number of plaques was counted.

**Antiviral multicycle growth assay**. Coronaviruses (SARS-CoV-2, MERS-CoV, and SARS-CoV), influenza viruses (H1N1 and H7N9 virus) and rhinovirus (0.005 MOI) were premixed with P9R or BSA (50–100 µg ml$^{-1}$) in PB for 1 h and then infected VERO-E6, MDCK, and RD cells, correspondingly. After 1 h infection, infectious media were removed and fresh media with supplemented P9R or BSA (50–100 µg ml$^{-1}$) were added to infected cells for virus and cell culture. At 24–30 h post infection, the supernatants of cells were collected for plaque assay.

**Cytotoxicity assay**. Cytotoxicity of peptides was determined by the detection of 50% cytotoxic concentration (CC$_{50}$) using a tetrazolium-based colorimetric MTT assay[13]. Briefly, cells were seeded in 96-well cell culture plate at an initial density of $2 \times 10^4$ cells per well in MEM or DMEM supplemented with 10% FBS and incubated for overnight. Cell culture media were removed and then DMEM supplemented with various concentrations of peptides and 1% FBS were added to each well. After 24 h incubation at 37 °C, MTT solution (5 mg ml$^{-1}$, 10 µl per well) was added to each well for incubation at 37 °C for 4 h. Then, 100 µl of 10% SDS in 0.01 M HCl was added to each well. After further incubation at room temperature with shaking overnight, the plates were read at OD570 using VictorTM X3 Multilabel Reader (PerkinElmer, USA). Cell culture wells without peptides were used as the experimental control and medium only served as a blank control.

**Transmission electron microscopy assay**. To determine the effect of P9R on viral particles, SARS-CoV-2 was pretreated by 100 µg ml$^{-1}$ of P9R or P9SHN for 1 h. The virus was fixed by formalin for overnight and then applied to continuous carbon grids. The grids were transferred into 4% uranyl acetate and incubated for 1 min. After removing the solution, the grids were air-dried at room temperature. For each peptide/DNA nanoparticle, three independent experiments were done for taking TEM images by FEI Tecnal G2-20 TEM.

**Hemolysis and hemolysis inhibition assay**. Serially diluted peptides in PBS were incubated with chicken red blood cells for 1 h at 37 °C. PBS was used as a 0% lysis control and 0.1% Triton X-100 as 100% lysis control. Plates were centrifuged at $350 \times g$ for 3 min to pellet non-lysed red blood cells. Supernatants used to measure hemoglobin release were detected by absorbance at 450 nm[47]. For hemolysis inhibition assay, P9R (200 µg ml$^{-1}$) or Arbidol (100 µg ml$^{-1}$) were mixed with or without same volume of H1N1 virus (HA titer >128) for 1 h, and then 60 µL of 2% chicken red blood cells was added for 15 min. PBS and Triton-100 (0.1%) were included as the negative and positive control of hemolysis. The precipitated erythrocytes were incubated with sodium citrate solution (pH of 4.9) for 25 min. The hemoglobin release in supernatants was detected at 450 nm.

**Peptide-virus binding assay**. Peptides (1.0 µg per well) dissolved in H$_2$O were coated onto ELISA plates and incubated at 4 °C overnight. Then, 2% BSA was used to block plates at 4 °C overnight. For virus binding to peptides, viruses were diluted in PB and then were added to ELISA plate for binding to the coated peptides at room temperature for 1 h. After washing the unbinding viruses, the binding viruses were lysed by RLT buffer of RNeasy Mini Kit (Qiagen, Cat# 74106) for viral RNA extraction. Viral RNA copies of binding viruses were measured by RT-qPCR.

**ELISA assay**. For ELISA assay[14], Peptides (1.0 µg per well) dissolved in H$_2$O were coated onto ELISA plates and incubated at 4 °C overnight. Then, 2% BSA was used to block plates at 4 °C overnight. For HA and S binding, 150 ng HA1 or S in solution I buffer (Sino Biological Inc., Cat# 11055-V08H4) was incubated with peptides at 37 °C for 1 h. The binding abilities of peptides to HA1 or S proteins were determined by incubation with rabbit anti-His-HRP (Invitrogen, Cat# R93125, 1: 2,000) at room temperature for 30 min. The reaction was developed by

adding 50 μl of TMB single solution (Life Technologies, Cat# 002023) for 15 min at 37 °C and stopped with 50 μl of 1 M $H_2SO_4$. Readings were obtained in an ELISA plate reader (Victor 1420 Multilabel Counter; PerkinElmer) at 450 nm.

**Viral RNA extraction and RT-qPCR.** Viral RNA was extracted by Viral RNA Mini Kit (QIAGEN, Cat# 52906, USA) according to the manufacturer's instructions. Real-time RT-qPCR was performed as we described previously[14]. Extracted RNA was reverse transcribed to cDNA using PrimeScript II 1st Strand cDNA synthesis Kit (Takara, Cat# 6210A) with GeneAmp® PCR system 9700 (Applied Biosystems, USA). The cDNA was then amplified using specific primers (Supplementary Table 2) for detecting SARS-CoV-2, MERS-CoV, SARS-CoV, H1N1, H7N9, and rhinovirus using LightCycle® 480 SYBR Green I Master (Roach, USA). For quantitation, 10-fold serial dilutions of standard plasmid equivalent to $10^1$–$10^6$ copies per reaction were prepared to generate the calibration curve. Real-time qPCR experiments were performed using LightCycler® 96 system (Roche, USA).

**Endosomal acidification assay.** Endosomal acidification was detected with a pH-sensitive dye (pHrodo Red dextran, Invitrogen, Cat#P10361) according to the manufacturer's instructions as previously described but with slight modification[14]. First, MDCK cells were treated with BSA (25.0 μg ml$^{-1}$), P9 (25.0 μg ml$^{-1}$), P9R (25.0 μg ml$^{-1}$), PA1 (25.0 μg ml$^{-1}$), or P9RS (25.0 μg ml$^{-1}$) at 4 °C for 15 min. Second, MDCK cells were added with 100 μg ml$^{-1}$ of pH-sensitive dye and DAPI and then incubated at 4 °C for 15 min. Before taking images, cells were further incubated at 37 °C for 15 min and then cells were washed twice with PBS. Finally, PBS was added to cells and images were taken immediately with confocal microscope (Carl Zeiss LSM 700, Germany).

**Colocalization assay of peptide binding to virus in cells.** H1N1 virus was labeled by green Dio dye (Invitrogen, Cat#3898) according to the manufacture introduction. DIO-labeled virus was treated by TAMRA-labeled P9R and TAMRA-labeled P9RS for 1 h at room temperature. Pre-cool MDCK cells were infected by the peptide-treated virus on ice for 15 min and then moved to 37 °C for incubation for 15 min. Cells were washed twice by PBS and then fixed by 4% formalin for 1 h. Nuclei were stained by DAPI for taking images by confocal microscope (Carl Zeiss LSM 700, Germany).

**Nucleoprotein (NP) immunofluorescence assay.** According to our previous experiment[14], MDCK cells were seeded on cell culture slides and were infected with A(H1N1)pdm09 virus at 1 MOI pretreated with BSA (25.0 μg ml$^{-1}$), bafilomycin A1 (50.0 nM) or P9R (25.0 μg ml$^{-1}$). After 3.5 h post infection, cells were fixed with 4% formalin for 1 h and then permeabilized with 0.2% Triton X-100 in PBS for 5 min. Cells were washed by PBS and then blocked by 5% BSA at room temperature for 1 h. Cells were incubated with mouse IgG anti-NP (Millipore, Cat# 2817019, 1:600) at room temperature for 1 h and then washed by PBS for next incubation with goat anti-mouse IgG Alexa-488 (Life Technologies, Cat# 1752514, 1:600) at room temperature for 1 h. Finally, cells were washed by PBS and stained with DAPI. Images were taken by confocal microscope (Carl Zeiss LSM 700, Germany).

**NMR structure analysis of P9R.** Freshly prepared 1 mg ml$^{-1}$ (0.29 mM) of P9R in 0.5 ml solvent was used for the NMR study. Data were collected in $H_2O/D_2O$ (19:1 v/v), as well as 99.996% $D_2O$, with the internal reference trimethylsilyl-propanoic acid. All NMR spectra were acquired on either a Bruker AVANCE III 600 MHz spectrometer (Topspin 3.1, Bruker BioSpin, Germany) or a Bruker AVANCE III 700 MHz spectrometer at 25 °C. 2D $^1H$-$^1H$ correlation spectroscopy (COSY), total correlated spectroscopy (TOCSY) and nuclear Overhauser effect spectroscopy (NOESY) spectra were recorded for resonance assignments. Supplementary Fig. 9a displayed the fingerprint region of 2D TOCSY spectrum showing amide cross peak region of peptide P9R and Supplementary Fig. 9b showed the 2D NOESY spectrum of peptide P9R, at a mixing time of 300 ms. Inter-proton distance restraints were derived from 2D NOESY spectrum with mixing times of 300 ms and 500 ms using automated NOE assignment strategy followed by a manual check. NOE intensities and chemical shifts were extracted using CCPNMR Analysis 2.4.2[48] and served as inputs for the Aria program. Dihedral angle is predicted from the chemical shifts using the program DANGLE[49]. The NMR solution structure of P9R was calculated iteratively using Aria 2.3 program[50]. One hundred random conformers were annealed using distance restraints in each of the eight iteratively cycles of the combined automated NOE assignments and structure calculation algorithm. The final upper limit distance constraints output from the last iteration cycle were subjected to a thorough manual cross-checking and final water solvent structural refinement cycle. The 20 lowest energy conformers were retained from these refined 100 structures for statistical analysis. The convergence of the calculated structures was evaluated using root-mean-square deviations (RMSDs) analyses. The distributions of the backbone dihedral angles (φ, ψ) of the final converged structures were evaluated by representation of the Ramachandran dihedral pattern using PROCHECK-NMR[51]. Visualization of three-dimensional structures and electrostatic surface potential of P9R were achieved using UCSF Chimera 1.13.1[52]. The final water solvent refined structures were deposited at Protein Data Bank (https://www.rcsb.org/) under the code of 6M56 and the resonance assignments of P9R were

deposited at Biological Magnetic Resonance Bank (http://www.bmrb.wisc.edu) under the accession number of 36321.

**Antiviral analysis of P9R in mice.** BALB/c female mice, 10–12 weeks old, were kept in biosafety level 2 laboratory (housing temperature between 22 °C and 25 °C with dark/light cycle) and given access to standard pellet feed and water ad libitum. All experimental protocols followed the standard operating procedures of the approved biosafety level 2 animal facilities. Animal ethical regulations were approved by the Committee on the Use of Live Animals in Teaching and Research of the University of Hong Kong[46]. The mouse adapted H1N1 virus was used for lethal challenge of mice. To evaluate the therapeutic effect, mice were challenged with 3 $LD_{50}$ of the virus and then intranasally inoculated with PBS, P9, P9R, PA1 or zanamivir at six hours after the viral inoculation. Two more doses were given to H1N1-challenged mice at the following one day. Survival and general conditions were monitored for 16 days or until death. Viral loads in mouse lungs were measured at day 4 post infection.

**Statistical analysis.** Survival of mice and the statistical significance were analyzed by Gehan–Breslow–Wilcoxon test. The statistical significance of the other results was calculated by the two-tailed Student $t$ test. Results were considered significant at $P < 0.05$.

**Reporting summary.** Further information on research design is available in the Nature Research Reporting Summary linked to this article.

## Data availability

All data that support the conclusions of the study are available from the corresponding author upon request. NMR data were deposited at Protein Data Bank (https://www.rcsb.org/) under the code of 6M56 and Biological Magnetic Resonance Bank (http://www.bmrb.wisc.edu) under the accession number of 36321. Source data are provided with this paper.

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

## Acknowledgements

This study was partly supported by the donations of Richard Yu and Carol Yu, Michael Seak-Kan Tong, May Tam Mak Mei Yin, the Shaw Foundation Hong Kong, Respiratory Viral Research Foundation Limited, Hui Ming, Hui Hoy and Chow Sin Lan Charity Fund Limited, Chan Yin Chuen Memorial Charitable Foundation, Marina Man-Wai Lee, the Hong Kong Hainan Commercial Association South China Microbiology Research Fund, the Jessie & George Ho Charitable Foundation, Perfect Shape Medical Limited, and Kai Chong Tong; and funding from Health@InnoHK (Centre for Virology, Vaccinology and Therapeutics), Innovation and Technology Commission, The Government of the Hong Kong Special Administrative Region of the People's Republic of China, and the National Program on Key Research Project of China (grant no. 2020YFA0707500 and 2020YFA0707504). The funding sources had no role in the study design, data collection, analysis, interpretation, or writing of the report.

## Author contributions

H.Z. and K.Y. designed this study. H.Z., K.S., T.Y., M.B., H.L., C.L., and H.C. performed experiments. H.Z., K.T., K.Y. interpreted the findings. H.Z., K.T., K.S., M.Y., and K.Y. revised the paper.

## Competing interests

The authors declare no competing interests.
