## [Peer Review File · Nature Communications]

Peer Review File - Reviewers' comments first round

Reviewer #1 (Remarks to the Author):

Zhao and co-workers show data on the alkaline peptide P9R with broad antiviral activity against the novel SARS-CoV-2, two other highly pathogenic CoV (MERS-CoV and SARS-CoV), influenza viruses and non-enveloped rhinovirus. Overall, the work is sound. Experiments are carefully conducted and results are clearly presented with reasonable discussion. The 2019 new coronavirus has spread to the world in many countries, and there is an urgent need to develop broad-spectrum antivirals for treating novel emerging viruses from now on.

In this study, the authors are providing a novel concept of alkaline antiviral peptide. P9R could bind to different viruses and then inhibit host endosomal acidification. As far as I know, this is the first antiviral peptide which targets both of virus and host to exert very broad antiviral activity. The authors clearly showed that P9R could bind to different viruses and then could efficiently inhibit virus replication with low IC50. They used two control peptide PA1 and P9RS to indicate that the binding to virus only (PA1) or inhibiting host endosomal acidification only (P9RS) could not show antiviral activity. They used different methods to confirm the mechanism that the binding to viruses and the inhibition on host endosomal acidification are both needed for the antiviral activity of alkaline P9R. The results are solid supported by the experiments. The in vivo results indicated the efficiency of P9R for anti-influenza virus in mice. The low possibility of P9R inducing drug-resistance virus indicated that this broad-spectrum antiviral peptide, in principle, may have some clinical advantages because of the quickly emergency of drug-resistance virus during treatment.

Comments:

- Why did the authors only study respiratory viruses? There are other non-respiratory viruses which are pH-dependent viruses. Does P9R also have antiviral activity against other viruses? Have the authors tried other non-respiratory viruses?
- Why can P9R bind to different viruses? It seems that cysteines are important for defensin peptide. The authors may discuss this a little bit more in discussion.
- In line 121, authors made a peptide P9RS with multiple substitutions. As a result, P9RS lost the antiviral activity, but with the same positive charge as P9R. It is a good control for this study. How did the authors design this peptide P9RS?
- In line 217, 'enterovirus-A7' should be 'enterovirus-A71'
- Recently, the clinically approved drug Hydroxychloroquine was identified to be effective in inhibiting SARS-CoV-2. 'Hydroxychloroquine, a less toxic derivative of chloroquine, is effective in inhibiting SARS-CoV-2 infection in vitro'. Please discuss this in the discussion related to the chloroquine part.
- In line 559 of figure legend 4, 'Zana' is not necessary?
- Figure 5a: It is suggested to show the concentration of both Zanamivir and P9R as nM.

Reviewer #2 (Remarks to the Author):

The manuscript from Yuen et al., entitled "A broad-spectrum virus- and host-targeting antiviral peptide against SARS-CoV-2 and other respiratory viruses" reports on a broadly antiviral peptide, which has been derived from mouse β -beta defensin 4. The authors performed an extensive study against several viral strains, which revealed that the previously reported lead peptide P9 (Yuen et al., 2016, Scientific reports) can be converted in a peptide (P9R) with higher antiviral activity, if four amino acids at the C-terminal region have been replaced by positively charged arginine. The authors claim that the peptide acts upon binding to the viral surface and by inhibiting endosomal

acidification. Although there is currently an emerging need for new antivirals especially against CoVid-19 (SARS-CoV-2), and the principle strategy to develop naturally inspired peptides with antiviral potency is a promising approach I noticed several significant weaknesses in the paper. Meaning more control experiments and clarifications are needed to provide more evidence on the mechanism of action of the peptide, and to show that the peptide is applicable in a therapeutic setup.

Major concerns:

1. Peptides:

A) The solvent of the peptide needs to be stated. As the water solubility of the peptides is very low I was wondering which vehicle the authors used in their experiments, especially in vivo. This is also relevant in terms of the applicability of the peptide P9R.

B) In Figure 1a the authors mentioned "positive charges" of the peptides. To which pH the given values for the "net charge" are related to. How does it change during endosomal acidification?

C) The manuscript gives the impression that the positively charged (basic) peptides act buffering within endosomes. See page 6, line 115: "the degree of positive charge was correlated with the degree of inhibition of endosomal acidification". However, I cannot follow the hypothesis of P9R as a buffering peptide from a theoretical point of view. The pKa value of the guanidinium group of arginine is around 12, which means guanidinium is already protonated at physiological pH 7.4. These residues cannot buffer more protons. Histidines instead would act buffering at pH 4.5-6.5 in the endosomes. Can the authors comment on that. I suspect the sequence feature of P9R, provides the peptide another functionality, which brings me to the next point.

D) With the introduction of positive charges at the C-terminus of the peptide the authors created an amphiphilic peptide. It is highly probable that peptide P9R now disturbs the integrity of lipid membranes. Thus, it is possible that P9R acts virucidal, which need to be shown (see below) and discussed in the manuscript. In addition, the peptide could lead to proton leakage at the endosomal membrane and thus lead to a delay or even inhibition of acidification. However, I see that this point would need more extensive cell culture studies. However, virus fusion inhibition assays (e.g. virus induced hemolysis inhibition or fluorescence dequenching assays) with H1N1 virus are simple to setup and could give a first indication of the mode of action of the peptides (e.g. pH buffering). The same holds true for a potential virucidal inhibition: It would be highly relevant to show if peptide P9R is rupturing the virus integrity. Either by electron microscopy of virus particles treated with P9R or biochemically by virus incubation with peptide followed by dialysis. If the binding is reversible the virus titer should not decrease.

E) In all in vitro assays with virus, the authors first incubated virus with peptides and then treated cells with the virus-peptide mix. However, this approach reflects only a prophylactically antiviral treatment. It would be necessary to show at least with one virus a therapeutic setup in which cells get first infected before the treatment takes place. In the best case with a virus growth curve. If the peptides do not bind unspecifically to lipid membranes, virus inhibition should also take place on virus progeny. This would also support and rationalize the in vivo studies, in which a therapeutic setup seems to work.

F) For the in vivo studies no virus titers are given. It is not clear how the peptide treatments work, even in the therapeutic setup. Otherwise no direct correlation between peptide treatment and antiviral effect (virus titer reduction) can be drawn.

G) Although the authors showed cytotoxicity data I think it is important to show the interaction of the peptides with lipid membranes. As the peptides act broadly antiviral it needs to be shown that the peptides do not bind unspecifically to hydrophobic surfaces in general. I suggest simple hemolysis assays with red blood cells at 37°C with the peptides, but other experiments (e.g. with liposomes) could also be implemented. Further the authors used a fluorescent peptide version. Can the authors provide images of cells treated with the fluorescent peptide only to show that there is no strong cell membrane binding?

Minor concerns:

- In several figures the authors did not provide the meaning of the error bars. Is it SEM or SD?

- No NMR spectra or data are given for figure 3g.

- In the abstract, the authors write that P9R exhibited antiviral activity against pH-dependent viruses for membrane fusion. However, non-enveloped viruses do not fuse with the endosomal membrane although a relevance of the pH reduction in the endosomes might play a role for virus uncoating. This needs to be rephrased.

- Figure 3h should be Figure 3g in the article

Reviewer #3 (Remarks to the Author):

In this study, Zhao et al. analyze the broad-spectrum activity of peptides against various respiratory viruses including SARS-CoV-2 and H7N9. The rationale of the antiviral peptides is that it interacts with the virus particle and inhibits endosomal acidification. This is a very elegant way to inhibit replication of viruses with an endosomal stage and thus provides –at least theoretically– a target for a broad range of viruses. The peptides are then evaluated in cell culture and later the mouse model for influenza was used. In general, the study is interesting and has promising components. However, as it stands, the study has some major weaknesses that should be addressed.

Major points:

#1. Figure 1-3: Antiviral assessment was performed in cell culture. However, antiviral effects are shown in percentage, which is not very significant given that any reduction below 1log with influenza viruses is not considered of biological relevance. The authors should show the data on “infectious particles” (e.g. p.f.u.) rather than copy number that is not very robust.

#2: Best hit peptide P9R should be tested on primary airway epithelia with all viruses. Peptide should be tested at various concentrations to evaluate its impact on virus replication kinetics. The results should be shown as p.f.u.. This is particularly important since the mouse model is only used to assess the antiviral activity of the peptide P9R against H1N1 influenza.

#3: Figure 4: The mouse data are not convincing. The authors used in some groups only 5 mice. The challenge dose used with 3xLD50 is very low for H1N1. Given the low animal number, even variations among biological replicates could alter survival rates. The authors should use n=10-20 per peptide/concentration. For a solid and convincing challenge 10xLD50 should be used. Additionally, virus titers in the lungs of treated and non-treated mice should be measured.

#4: Figure 4: In the absence of animal data on coronaviruses, the authors should at least repeat the mouse experiments with H7N9 to prove their claim of having an in vitro and in vivo potent antiviral peptide against various respiratory viruses.

Minor point:

The manuscript requires amendments in language and grammar.

Point-to-point response

Reviewers' comments:

Reviewer #1 (Remarks to the Author):

Zhao and co-workers show data on the alkaline peptide P9R with broad antiviral activity against the novel SARS-CoV-2, two other highly pathogenic CoV (MERS-CoV and SARS-CoV), influenza viruses and non-enveloped rhinovirus. Overall, the work is sound. Experiments are carefully conducted and results are clearly presented with reasonable discussion. The 2019 new coronavirus has spread to the world in many countries, and there is an urgent need to develop broad-spectrum antivirals for treating novel emerging viruses from now on.

In this study, the authors are providing a novel concept of alkaline antiviral peptide. P9R could bind to different viruses and then inhibit host endosomal acidification. As far as I know, this is the first antiviral peptide which targets both of virus and host to exert very broad antiviral activity. The authors clearly showed that P9R could bind to different viruses and then could efficiently inhibit virus replication with low IC50. They used two control peptide PA1 and P9RS to indicate that the binding to virus only (PA1) or inhibiting host endosomal acidification only (P9RS) could not show antiviral activity. They used different methods to confirm the mechanism that the binding to viruses and the inhibition on host endosomal acidification are both needed for the antiviral activity of alkaline P9R. The results are solid supported by the experiments. The in vivo results indicated the efficiency of P9R for anti-influenza virus in mice. The low possibility of P9R inducing drug-resistance virus indicated that this broad-spectrum antiviral peptide, in principle, may have some clinical advantages because of the quickly emergency of drug-resistance virus during treatment.

Comments:

- Why did the authors only study respiratory viruses? There are other non-respiratory viruses which are pH-dependent viruses. Does P9R also have antiviral activity against other viruses? Have the authors tried other non-respiratory viruses?

Response:

Thank you for this comment. There are non-respiratory viruses which also are pH dependent. In this study, we focused on respiratory viruses because we think that peptide P9R should be more effective and economic for topical administration against respiratory viruses. In the coming future, we will evaluate the antiviral activity of P9R or derivatives for non-respiratory viruses.

- Why can P9R bind to different viruses? It seems that cysteines are important for defensin peptide. The authors may discuss this a little bit more in discussion.

Response:

Thank you for your suggestion. We added these possibilities of the broadly binding mechanism of P9R in discussion paragraph from line 241 to line 246 ‘The broadly binding mechanism of P9R to different viral proteins may be due to the flexible structure of P9R with positively charged surface (Fig. 3g). The flexible structure may allow P9R to change its structure to fit targeting proteins for broad-specificity bindings, and the positive charge of P9R may play a role for binding to viruses with negatively charged surface. The five cysteines in P9R may also affect the structure-based binding because previous studies indicated that cysteine substitutions could affect defensin-peptide structure and biological activity.’ Peptide P9RS lost the binding ability to viruses, which might indicate that cysteines are important for the structure-based binding. More co-binding structure study will be needed to determine the binding mechanism.

- In line 121, authors made a peptide P9RS with multiple substitutions. As a result, P9RS lost the antiviral activity, but with the same positive charge as P9R. It is a good control for this study. How did the authors design this peptide P9RS?

Response:

Thank you for this question. We originally substituted five cysteines to serines in the sequence. In order to get the same positive charge as P9R, we finally selected P9RS which did not bind to virus and have the same positive charge as P9R.

- In line 217, ‘enterovirus-A7’ should be ‘enterovirus-A71’

Response:

We apologize for the mistake. We have now corrected the virus name in the revised manuscript.

- Recently, the clinically approved drug Hydroxychloroquine was identified to be effective in inhibiting SARS-CoV-2. ‘Hydroxychloroquine, a less toxic derivative of chloroquine, is effective in inhibiting SARS-CoV-2 infection in vitro’. Please discuss this in the discussion related to the chloroquine part.

Response:

Thank you for this comment. We have now added the discussion of the chloroquine and hydroxychloroquine in the Discussion section from line 226 to line 234 as followings: ‘In previous studies, the clinically approved anti-malarial drug chloroquine and hydroxychloroquine with activity of inhibiting endosomal acidification had been demonstrated to inhibit enterovirus-A71, zika virus and SARS-CoV-2.’ ‘However, researchers demonstrated the lack of protection of chloroquine *in vivo* for treating influenza virus and Ebola virus. Differing from these drugs by interfering host endosomal acidification without targeting viruses, P9R inhibits viral replication by binding to viruses and then inhibiting host acidification of the endosome containing the virus bound by P9R, which allows P9R to selectively and efficiently inhibit the replication cycle of endosomal viruses.’

- In line 559 of figure legend 4, 'Zana' is not necessary?

Response:

Thank you for this correction. We corrected it accordingly

- Figure 5a: It is suggested to show the concentration of both Zanamivir and P9R as nM.

Response:

Thank you for this comment. We added the nM for zanamivir and μ M for P9R in the Fig. 5a. If we use nM for P9R as the unit, the digital number is very long. We hope that you will agree with us to add μ M for P9R.

Reviewer #2 (Remarks to the Author):

The manuscript from Yuen et al., entitled “A broad-spectrum virus- and host-targeting antiviral peptide against SARS-CoV-2 and other respiratory viruses” reports on a broadly antiviral peptide, which has been derived from mouse β -beta defensin 4. The authors performed an extensive study against several viral strains, which revealed that the previously reported lead peptide P9 (Yuen et al., 2016, Scientific reports) can be converted in a peptide (P9R) with higher antiviral activity, if four amino acids at the C-terminal region have been replaced by positively charged arginine. The authors claim that the peptide acts upon binding to the viral surface and by inhibiting endosomal acidification. Although there is currently an emerging need for new antivirals especially against CoVid-19 (SARS-CoV-2), and the principle strategy to develop naturally inspired peptides with antiviral potency is a promising approach I noticed several significant weaknesses in the paper. Meaning more control experiments and clarifications are needed to provide more evidence on the mechanism of action of the peptide, and to show that the peptide is applicable in a therapeutic setup.

Major concerns:

1. Peptides:

A) The solvent of the peptide needs to be stated. As the water solubility of the peptides is very low I was wondering which vehicle the authors used in their experiments, especially in vivo. This is also relevant in terms of the applicability of the peptide P9R.

Response:

Thank you for reviewer's suggestion. The peptide P9R is dissolved in water for this study. The solubility of P9R in water can be higher than 5 mg/ml. The information is now stated in the Methods section in line 271: “All peptides were dissolved in water. The solubility of P9R in water is greater than 5 mg/ml”.

B) In Figure 1a the authors mentioned “positive charges” of the peptides. To which pH the given values for the “net charge” are related to. How does it change during endosomal acidification?

Response:

Thank you for these questions. The net positive charges are related to pH 7.0. We have now stated this in the figure legend of Fig. 1a (line 534). The positive charge of P9R will be increased from +5.3 to +6.0 when the pH is changed from pH 7.5 to pH 6.0 during the acidification according to the analysis (<http://protcalc.sourceforge.net/>).

C) The manuscript gives the impression that the positively charged (basic) peptides act buffering within endosomes. See page 6, line 115: “the degree of positive charge was correlated with the degree of inhibition of endosomal acidification”. However, I cannot follow the hypothesis of P9R as a buffering peptide from a theoretical point of view. The pKa value of the guanidinium group of arginine is around 12, which means guanidinium is already protonated at physiological pH 7.4. These residues cannot buffer more protons. Histidines instead would act buffering at pH 4.5-6.5 in the endosomes. Can the authors comment on that. I suspect the sequence feature of P9R, provides the peptide another functionality, which brings me to the next point.

Response:

Thank you for the comments. The pKa of the guanidinium group of arginine is ~12 at neutral pH. However, the positive charge of P9R will be changed from +5.3 to +6.0 when the pH is changing from pH 7.5 to pH 6.0 according to the analysis (<http://protcalc.sourceforge.net/>), which means that with pH decreasing from 7.5 to 6.0, P9R could buffer more proton (H^+) in endosomes. Furthermore, we think that the positively charged P9R⁺ (+6.0) will have a negative effect on the pumping of positive proton (H^+) into endosomes because of the charge-charge repulsion. Positive-charged basic P9R is likely having a similar mechanism of action as chloroquine, which is a well-known weak base inhibiting viral infection through raising endosomal pH [Savarino 2003]. Indeed, in Fig. 2ab, we demonstrated the inhibition of P9R on endosomal acidification in MDCK cells. We have modified the text in discussion to clarify our findings as shown in line 223: ‘In this study, with the increased positive charge in P9R, it could more efficiently inhibit pH-dependent viruses than that of P9. The more positive charge in P9R allowed the peptide to more efficiently reduce protons inside endosomes, and thereby inhibiting the endosomal acidification.’

Reference: Andrea Savarino, Johan R Boelaert, Antonio Cassone, Giancarlo Majori, and Roberto Cauda. Effects of chloroquine on viral infections: an old drug against today’s diseases? *Lancet Infect Dis* 2003; 3: 722–27.

D) With the introduction of positive charges at the C-terminus of the peptide the authors created an amphiphilic peptide. It is highly probable that peptide P9R now disturbs the integrity of lipid membranes. Thus, it is possible that P9R acts virucidal, which need to be shown (see below) and discussed in the manuscript. In addition, the peptide could lead to proton leakage at the endosomal membrane and thus lead to a delay or even inhibition of acidification. However, I see that this point would need more extensive cell culture studies. However, virus fusion inhibition assays (e.g. virus induced hemolysis inhibition or fluorescence dequenching assays) with H1N1 virus are simple to setup and could give a first indication of the mode of action of the peptides

(e.g. pH buffering). The same holds true for a potential virucidal inhibition: It would be highly relevant to show if peptide P9R is rupturing the virus integrity. Either by electron microscopy of virus particles treated with P9R or biochemically by virus incubation with peptide followed by dialysis. If the binding is reversible the virus titer should not decrease.

Response:

Thank you for the suggestions. We have performed several additional experiments as recommended by the reviewer. First, we used transmission electron microscopy to show that P9R did not disrupt SARS-Co-2 virus particles (the new Supplementary Fig. 2 in the revised manuscript in line 116). Second, we showed that P9R-treated virus could still attach to host cell surface similarly to non-treated virus (new Supplementary Fig. 5). Third, we demonstrated that P9R did not inhibit virus-host cell membrane fusion by hemolysis inhibition assay for influenza A(H1N1) (the new Supplementary Fig. 6 in the revised manuscript). Therefore, P9R did not affect virus-host membrane fusion. Fourth, we showed that the antiviral activity of P9R was irreversible (new Supplementary Fig. 7). After extremely dilution of P9R binding to virus, P9R could significantly inhibit viral infection at 0.05 µg/ml or 0.01 µg/ml, which were >10-fold lower than the IC₅₀ of P9R. The irreversible antiviral activity of P9R at the low concentration (0.01~0.05 µg/ml) indicated that the antiviral activity of P9R did not mainly rely on targeting host to disrupt host lipid membrane. Furthermore, since P9R inhibited the replication of rhinovirus (Fig. 1f), which does not have lipid membrane, it is unlikely that disrupting the lipid membrane is the main mechanism of action of P9R. Taken together, our additional experiments confirmed that P9R did not disrupt the virus particle, prevent attachment, or inhibit virus-host cell membrane fusion. The antiviral activity of P9R was irreversible and did not rely on disrupting lipid membrane. Results and discussion were shown in manuscript from line 116 to line 123: ‘In order to investigate whether P9R could have antiviral action before endosomal acidification, we showed that P9R did not disrupt viral particles under TEM analysis (Supplementary Fig. 2), did not cause hemolysis of RBC (Supplementary Fig. 3) and did not show antiviral activity when cells were pretreated by P9R before viral infection (Supplementary Fig. 4). Furthermore, P9R did not affect viral attachment (Supplementary Fig. 5) and did not inhibit viral fusion by hemolysis inhibition assay (Supplementary Fig. 6). The antiviral activity of P9R was irreversible after P9R binding to virus (Supplementary Fig. 7). Results indicated that the irreversible antiviral activity of P9R did not rely on disrupting lipid membrane or inhibit virus-host fusion by directly binding’.

E) In all in vitro assays with virus, the authors first incubated virus with peptides and then treated cells with the virus-peptide mix. However, this approach reflects only a prophylactically antiviral treatment. It would be necessary to show at least with one virus a therapeutic setup in which cells get first infected before the treatment takes place. In the best case with a virus growth curve. If the peptides do not bind unspecifically to lipid membranes, virus inhibition

should also take place on virus progeny. This would also support and rationalize the in vivo studies, in which a therapeutic setup seems to work.

Response:

Thank you very much for this comment. We have now performed the experiment as recommended by the reviewer. We added P9R to infected cells at 6 h post-infection and tested the growth curve of treated- and untreated viruses. As shown in new Supplementary Fig. 1 (in line 96), P9R could significantly reduce viral replication at 24 h and 30 h post-infection when compared with untreated virus.

F) For the in vivo studies no virus titers are given. It is not clear how the peptide treatments work, even in the therapeutic setup. Otherwise no direct correlation between peptide treatment and antiviral effect (virus titer reduction) can be drawn.

Response:

Thank you for this comment. We have now measured the lung viral loads at day 4 post-infection (new Fig. 4c and Fig. 4f in line 185 and line 189). P9R could significantly inhibit viral replication in mouse lungs when compared with untreated mice.

G) Although the authors showed cytotoxicity data I think it is important to show the interaction of the peptides with lipid membranes. As the peptides act broadly antiviral it needs to be shown that the peptides do not bind unspecifically to hydrophobic surfaces in general. I suggest simple hemolysis assays with red blood cells at 37°C with the peptides, but other experiments (e.g. with liposomes) could also be implemented. Further the authors used a fluorescent peptide version. Can the authors provide images of cells treated with the fluorescent peptide only to show that there is no strong cell membrane binding?

Response:

Thank you for these comments. We have performed the experiment suggested by the reviewer. We incubated the red blood cells with peptide as shown in the new Supplementary Fig. 3 (in line 118). P9R did not cause hemolysis even at high concentration of 100 µg/ml. When cells were pretreated by P9R, P9R did not show antiviral activity against influenza A(H1N1) virus infection (new Supplementary Fig. 4 in line 119). As shown in Fig. 2e, there was no obvious binding of P9R to cell membrane when cells were treated by TAMRA-labeled P9R. Hence our experiments indicated that the antiviral activity of P9R did not rely on targeting lipid membrane.

Minor concerns:

- In several figures the authors did not provide the meaning of the error bars. Is it SEM or SD?

Response:

Thanks for this comment. We revised the figure legends to state that the error bars are SD.

- No NMR spectra or data are given for figure 3g.

Response:

Thanks for this comment. We have deposited the NMR data in PDB ID 6M56 (<https://www.rcsb.org/>) and BMRB ID 36321 (<http://www.bmrwisc.edu>), which is described in line 394 of Method section. P9R NMR spectra were provided in new Supplementary Fig. 9.

- In the abstract, the authors write that P9R exhibited antiviral activity against pH-dependent viruses for membrane fusion. However, non-enveloped viruses do not fuse with the endosomal membrane although a relevance of the pH reduction in the endosomes might play a role for virus uncoating. This needs to be rephrased.

Response:

Thanks for this correction. We have corrected the statement in the abstract to ‘Here, we showed that a defensin-like peptide P9R exhibited potent antiviral activity against pH-dependent viruses that require endosomal acidification for virus infection...’ in line 29.

- Figure 3h should be Figure 3g in the article

Response:

Thanks for this correction. We had corrected it to Fig. 3g in line 242.

Reviewer #3 (Remarks to the Author):

In this study, Zhao et al. analyze the broad-spectrum activity of peptides against various respiratory viruses including SARS-CoV-2 and H7N9. The rationale of the antiviral peptides is that it interacts with the virus particle and inhibits endosomal acidification. This is a very elegant way to inhibit replication of viruses with an endosomal stage and thus provides –at least theoretically- a target for a broad range of viruses. The peptides are then evaluated in cell culture and later the mouse model for influenza was used. In general, the study is interesting and has promising components. However, as it stands, the study has some major weaknesses that should be addressed.

Major points:

#1. Figure 1-3: Antiviral assessment was performed in cell culture. However, antiviral effects are shown in percentage, which is not very significant given that any reduction below 1log with

influenza viruses is not considered of biological relevance. The authors should show the data on “infectious particles” (e.g. p.f.u.) rather than copy number that is not very robust.

Responses:

Thank you very much for this comment. We have now showed the data of viral PFU in new Fig. 1i. In Fig. 2d, the relative viral RNA copies represented the amount of virus binding to peptides. They were viral lysates. In Fig. 3b, the relative viral RNA copies represented the amount of virus binding to peptides. They were viral lysates. No viral PFU could be detected.

#2: Best hit peptide P9R should be tested on primary airway epithelia with all viruses. Peptide should be tested at various concentrations to evaluate its impact on virus replication kinetics. The results should be shown as p.f.u. This is particularly important since the mouse model is only used to assess the antiviral activity of the peptide P9R against H1N1 influenza.

Response:

Thank you for the comment. In this study, we mainly introduced a new concept of antiviral peptide targeting both virus and host to show the broad spectrum of antiviral activities. Thus, we selected MDCK, Vero E6 and RD cell lines that are commonly used for the evaluation of antivirals (Wenhao Dai, et al. Structure-based design of antiviral drug candidates targeting the SARS-CoV-2 main protease, *Science*, 2020¹; Manli Wang, et al, Remdesivir and chloroquine effectively inhibit the recently emerged novel coronavirus (2019-nCoV) in vitro, *Cell Res*, 2020²; David J. Holthausen, et al, An Amphibian Host Defense Peptide Is Virucidal for Human H1 Hemagglutinin-Bearing Influenza Viruses, *Immunity*, 2017³; Y.W. Tan, et al, Antiviral activities of peptide-based covalent inhibitors of the Enterovirus 71 3C protease, *Sci Rep*, 2016⁴)

#3: Figure 4: The mouse data are not convincing. The authors used in some groups only 5 mice. The challenge dose used with 3xLD50 is very low for H1N1. Given the low animal number, even variations among biological replicates could alter survival rates. The authors should use n=10-20 per peptid/concentration. For a solid and convincing challenge 10xLD50 should be used. Additionally, virus titers in the lungs of treated and non-treated mice should be measured.

Response:

Thank you for this comment. According to reviewer’s suggestion, we tested more mice in each group. We investigated the survival data by including 10-20 mice in each group for mice treated by P9R, P9, zanamivir, PA1, and PBS (Fig. 4a and 4d in line 183 and line 189). And we measured the viral loads in mouse lungs at day 4 post infection (Fig. 4c and 4f in line 187 and line 188). The viral loads indicated that zanamivir and P9R could significantly reduce the viral replication when compared with the mice treated by PBS. In this study, we used intranasal inoculation with drug or PBS to do the treatment and we initially used 12xLD₅₀ to test the protection. However, in the 12xLD₅₀ condition, even zanamivir could only protect 20% mouse

survival. Thus, we selected 3xLD₅₀ as the 100% lethal model to evaluate the in vivo antiviral activity.

#4: Figure 4: In the absence of animal data on coronaviruses, the authors should at least repeat the mouse experiments with H7N9 to prove their claim of having an in vitro and in vivo potent antiviral peptide against various respiratory viruses.

Response:

Thank you for this comment. In this study, we are trying to introduce a novel antiviral concept of broad-spectrum antiviral peptide P9R, which targets both of virus and host to broadly inhibit viral infection. This P9R should be a good candidate for further development with more potent antiviral activity. We tried to demonstrate the in vivo antiviral activity of P9R by using H1N1-infected mice. In the coming future, we will try to develop new antiviral peptide with more potent antiviral activity and then will try to test the antiviral activity of the new potent antiviral peptide in mice with more viruses. We hope that by publishing our presently available data as a proof of concept at this stage would allow other groups to start testing the concept of broad-spectrum antiviral peptide targeting virus and host and therefore the discovery of better peptide variants for this purpose.

Minor point:

The manuscript requires amendments in language and grammar.

Response:

Thank you for this comment. We had asked academic staff of English speaking to check and amend the language.

1. Dai, W. et al. Structure-based design of antiviral drug candidates targeting the SARS-CoV-2 main protease. *Science* (2020).
2. Wang, M. et al. Remdesivir and chloroquine effectively inhibit the recently emerged novel coronavirus (2019-nCoV) in vitro. *Cell Res* (2020).
3. Holthausen, D.J. et al. An Amphibian Host Defense Peptide Is Virucidal for Human H1 Hemagglutinin-Bearing Influenza Viruses. *Immunity* **46**, 587-595 (2017).
4. Tan, Y.W. et al. Antiviral activities of peptide-based covalent inhibitors of the Enterovirus 71 3C protease. *Sci Rep* **6**, 33663 (2016).

Peer Review File - Reviewers' comments second round

Reviewer #2 (Remarks to the Author):

The authors improved the manuscript from Yuen et al. significantly and addressed all questions and comments with intensive efforts. From my view this article is now suitable for publication in nature communications.

Reviewer #3 (Remarks to the Author):

The authors have sufficiently addressed my major points. The revised manuscript has significantly improved. One minor point: the survival graphs should be presented as Kaplan-Meier plots.

Point-by-point response

REVIEWERS' COMMENTS:

Reviewer #2 (Remarks to the Author):

The authors improved the manuscript from Yuen et al. significantly and addressed all questions and comments with intensive efforts. From my view this article is now suitable for publication in nature communications.

Response:

Thanks for this comment.

Reviewer #3 (Remarks to the Author):

The authors have sufficiently addressed my major points. The revised manuscript has significantly improved. One minor point: the survival graphs should be presented as Kaplan-Meier plots.

Response:

Thanks for this comment. We have revised the survival graphs to the Kaplan-Meier plots.